# Small-Sample Authenticity Identification and Variety Classification of *Anoectochilus roxburghii* (Wall.) Lindl. Using Hyperspectral Imaging and Machine Learning

**DOI:** 10.3390/plants14081177

**Published:** 2025-04-10

**Authors:** Yiqing Xu, Haoyuan Ding, Tingsong Zhang, Zhangting Wang, Hongzhen Wang, Lu Zhou, Yujia Dai, Ziyuan Liu

**Affiliations:** 1College of Optical, Mechanical and Electrical Engineering, Zhejiang A&F University, Hangzhou 311300, China; xuyiqing136@zafu.edu.cn (Y.X.); dhytiaspetto@outlook.com (H.D.); 18656562509@163.com (T.Z.); 13511592738@163.com (Z.W.); zhoulu885@163.com (L.Z.); daiyujia@zafu.edu.cn (Y.D.); 2State Key Laboratory of Subtropical Silviculture, Department of Chinese Herbal Medicine Zhejiang A&F University, Hangzhou 311300, China; wang.hongzhen@163.com

**Keywords:** hyperspectral imaging, *Anoectochilus roxburghii* (Wall.) Lindl., authenticity identification, variety classification, machine learning

## Abstract

This study aims to utilize hyperspectral imaging technology combined with machine learning methods for the authenticity identification and classification of *Anoectochilus roxburghii* and its counterfeit species. Hyperspectral data were collected from the front and back leaves of nine species of Goldthread and two counterfeit species (Bloodleaf and Spotted-leaf), followed by classification using a variety of machine learning models, including Support Vector Machine (SVM), K-Nearest Neighbors (KNN), Random Forest (RF), Linear Discriminant Analysis (LDA), and Convolutional Neural Networks (CNN). The experimental results demonstrated that the SVM model achieved 100% classification accuracy for distinguishing Goldthread from its counterfeit species, effectively capturing the spectral differences between the front and back leaves. In contrast, traditional machine learning models showed varied performance, with SVM proving superior due to its ability to handle high-dimensional feature spaces. The introduction of a multi-view spectral fusion CNN model, which integrates spectral data from both the front and back leaves, further enhanced classification accuracy, achieving a perfect classification rate of 100%. This approach highlights the potential of hyperspectral imaging and machine learning in plant authenticity identification and provides a new perspective for the detection of counterfeit species.

## 1. Introduction

*Anoectochilus roxburghii* (Wall.) Lindl., commonly known as Goldthread, is a highly prized medicinal herb in traditional Asian medicine, renowned for its anti-inflammatory, antioxidant, and anticancer properties [1,2,3]. However, due to its high market demand, *A. roxburghii* is vulnerable to adulteration and misidentification [4,5,6]. This substitution not only undermines market integrity but also poses significant health risks due to the loss of therapeutic efficacy and potential toxicity. Ensuring the authenticity and precise classification of *A. roxburghii* is therefore critical for consumer safety and quality control in herbal medicine production.

Traditional methods for the authenticity identification and variety classification of medicinal plants primarily rely on visual inspection, microscopic examination, and chemical analysis [7]. In recent years, advanced techniques such as genomic analysis, mass spectrometry, and DNA testing have been developed to enhance identification accuracy [8,9,10,11]. These approaches are labor-intensive, destructive to samples, and often fail to distinguish subtle interspecies variations. Recent advancements in spectral technology have shown promise in addressing these challenges [12]. For instance, Li et al. [13] utilized near-infrared spectroscopy combined with chemometrics to establish a partial least squares discriminant analysis (PLS-DA) model, enabling the accurate classification of authentic *A. roxburghii* powder and its two counterfeit counterparts. Similarly, Chai et al. [14] employed near-infrared spectroscopy to obtain spectral data of *A. roxburghii* and its adulterants and designed an improved one-dimensional convolutional neural network (1D-CNN) model for processing the NIRS data, distinguishing between genuine and counterfeit *A. roxburghii*. Additionally, Li [15] proposed a fast and accurate classification model for *A. roxburghii* varieties based on a handheld near-infrared spectrometer and Adaboost ensemble learning, achieving an accuracy of up to 95.6%. These studies show that near-infrared spectroscopy offers high accuracy in the detection of *A. roxburghii*, but its application is typically limited to powdered forms of the plant, as it requires grinding, compressing, or other sample preparation methods.

Although previous studies [14,15] successfully distinguished intact *A. roxburghii* from its adulterants, they faced a key limitation: destructive sampling, which restricts their applicability for non-invasive plant authentication. Moreover, these methods inherently lose spatial and structural information, potentially overlooking biochemical heterogeneity between leaf surfaces. In contrast, hyperspectral imaging (HSI) provides a non-destructive alternative, capturing spatially resolved spectral features from both the adaxial and abaxial leaf surfaces of intact plants [16]. This approach not only preserves sample integrity but also enhances classification accuracy by extracting complementary biochemical information [17]. HSI, in particular, preserves sample integrity, captures a broad range of spectral bands across the electromagnetic spectrum, and provides detailed spectral information that can be used for precise material identification and quality assessment. HSI and machine learning (ML) are increasingly utilized in botanical research, as noted in the referenced review [18]. Applications include early detection of wheat fusarium head blight and grapevine powdery mildew, monitoring nitrogen deficiency in rice and drought stress in maize, distinguishing citrus cultivars and coffee bean varieties, and mapping soil organic matter for precision agriculture. These examples highlight the versatility of HSI-ML in plant science and sustainable agriculture. ML and deep learning (DL) have revolutionized pattern recognition in high-dimensional datasets [19,20], making them ideal for analyzing hyperspectral data [21,22,23,24]. While conventional ML models are widely used, their performance depends heavily on manual feature engineering and preprocessing. In contrast, DL architectures like convolutional neural networks autonomously extract discriminative features, enabling end-to-end classification. Despite these advancements, the potential of *A. roxburghii* for authenticity identification and multi-variety classification remains largely unexplored. Moreover, most studies rely on single-view spectral data (e.g., leaf front), overlooking the complementary insights provided by multi-view perspectives (e.g., front and back leaves). To date, no research has integrated multi-view HSI with hybrid ML/DL frameworks to simultaneously tackle the challenges of authenticity verification and intra-species classification in *A. roxburghii.*

This study bridges these gaps by proposing a novel framework that combines hyperspectral imaging with machine learning for non-destructive, high-precision identification of *A. roxburghii* and its counterfeit species. We introduce three key innovations: (1) Multi-view spectral fusion: Leveraging front and back leaf spectra to capture complementary biochemical and morphological features. (2) Hybrid ML/DL modeling: Comparing traditional ML algorithms (SVM, KNN, RF) with a custom 1D-CNN architecture optimized for spectral data. (3) Comprehensive validation: Evaluating model robustness across nine *A. roxburghii* varieties and two counterfeit species under diverse preprocessing techniques.

Our work not only advances the application of HSI in medicinal plant authentication but also provides a scalable, non-destructive solution for quality assurance in herbal markets. By addressing the limitations of existing methods, this research contributes to safeguarding consumer health and promoting sustainable practices in traditional medicine industries.

## 2. Materials and Methods

### 2.1. Sample Preparation

The *A. roxburghii* samples and counterfeit samples used in this experiment were provided and identified by Professor Wang Hongzhen from the Department of Traditional Chinese Herbal Medicine at Zhejiang A&F University. A total of 9 different varieties of *A. roxburghii* were included: Small Round Leaf (1), Pointed Leaf (2), Red Sunset (3), J6 Male (4), Colorful Sunset (5), Large Round Leaf (6), Red Sunset Large Leaf (7), Golden Vein No. 1 (8), and Taiwan Red (9). Two counterfeit species, Ludisia discolor (Ker-Gawl.) A. Rich. (L) and Goodyera schlechtendaliana Rchb. f. (G), were also included. Each variety included 10 leaves of similar size and maturity to minimize biological variability. Both the adaxial (front) and abaxial (back) sides of each leaf were scanned to capture multi-view spectral information. Samples were cleaned with distilled water to remove surface contaminants and air-dried under controlled laboratory conditions (25 °C, 60% humidity) prior to imaging.

### 2.2. Hyperspectral Image System

The hyperspectral imaging system used in this study is shown in Figure 1 (GaiaField-N17E, Dualix Spectral Imaging, Chengdu, China). It covers a spectral range from 900 nm to 1700 nm, with a spectral resolution of 5 nm and a spatial resolution of 640 pixels across 512 bands. The system consists of an indoor testing chamber equipped with four 50 W halogen lamps to provide stable illumination. The scanning hyperspectral spectrometer uses an array detector oriented perpendicular to the direction of motion, enabling it to scan two-dimensional space as the platform advances. The conveyor belt speed is set to 0.8 cm/s, ensuring that the samples pass through the scanning area at a stable and uniform speed. The exposure time is adjusted to auto, with the gain factor set to 1. The vertical distance between the sample and the lens is fixed at 42 cm. Samples are placed on a black base plate during scanning to enhance image contrast and recognition, as well as to minimize the interference of background diffuse reflection on the sample’s spectral data. During the data collection process, the hyperspectral camera scans each sample twice to improve data reliability and repeatability. After the scanning is completed, the raw hyperspectral images undergo black-and-white correction [25]. Subsequently, spectral extraction is performed for the regions of interest (ROI) of each sample using the software ENVI 5.3. In ENVI, a rectangular frame is used to randomly select a quarter of each sample as the ROI. Each pixel within the ROI contains a set of different spectral information. The final spectral value of the sample is obtained by averaging the spectral reflectance of all pixels within that region.

### 2.3. Data Preprocessing

In this study, various filtering techniques were applied to preprocess the hyperspectral data to reduce noise and enhance data quality, ensuring the accuracy and robustness of subsequent analyses. Since our study focuses on small-sample learning, we applied spectral preprocessing techniques such as denoising and augmentation to enhance the model’s robustness under limited data conditions. Additionally, cross-validation was employed to ensure reliable model evaluation and prevent overfitting. The following filtering methods were used:

Median filtering (MF). Median filtering is a non-linear filtering technique that replaces each pixel value with the median value of its neighborhood. This method is particularly effective in removing salt-and-pepper noise. The filtering process involves specifying the window size to determine the neighborhood for each pixel, then computing the median value within that neighborhood and replacing the original pixel value. Median filtering preserves edge details and is particularly useful for removing impulsive noise without blurring the image too much.

Average filtering (AF). Average filtering works by replacing each pixel value with the average value of its neighboring pixels. It is effective for reducing random noise but may blur edges, making it less effective in preserving fine details. This method is computationally simple and efficient, making it a popular choice for initial noise reduction in hyperspectral data.

Gaussian filtering (GF). Gaussian filtering is a common smoothing technique that uses a Gaussian function to weigh the neighboring pixel values, giving higher weights to those closer to the center. This method smooths the image while retaining more details in the central region compared to average filtering. Gaussian filtering is widely used in hyperspectral data preprocessing because it effectively reduces noise while minimizing the blurring effect.

Savitzky–Golay (SG) Filtering. The Savitzky–Golay filter is a smoothing method commonly used in signal processing. It applies polynomial fitting within a sliding window to smooth the data, thereby reducing high-frequency noise. In hyperspectral data processing, SG filtering helps smooth out small random fluctuations while preserving the main signal features.

Principal Component Analysis (PCA). PCA is an unsupervised statistical method for dimensionality reduction of data. It extracts the most important features, called principal components, by maximizing the variance of the data. In spectral analysis, PCA can help identify differences between samples and potential chemical changes.

### 2.4. Spectral Data Processing

Various classification models are used in this paper. The dataset was split using a stratified 80/20 ratio, ensuring that all classes were proportionally represented in both the training and testing sets. Additionally, a 5-fold cross-validation strategy was employed to assess the robustness of the models.

Support Vector Machine (SVM). SVM was employed for the classification of nine *A. roxburghii* varieties based on hyperspectral data. The SVM algorithm identifies the optimal hyperplane that maximizes the margin between different classes, offering strong generalization capabilities. The Radial Basis Function (RBF) kernel was selected due to its effectiveness in handling non-linear data patterns. Prior to training, the spectral data were standardized using StandardScaler to ensure consistency in feature scales. Hyperparameters, including the penalty parameter (C), kernel type, gamma, and polynomial degree, were optimized using Grid Search with 5-fold cross-validation. Model performance was evaluated using accuracy, precision, recall, F1-score, and confusion matrix, providing a comprehensive assessment of classification accuracy and errors.

K-Nearest Neighbors (KNN). The KNN algorithm was used for spectral classification, where the classification of a sample is based on the majority vote of its nearest neighbors in the feature space. The KNN algorithm first calculates the distance between the unknown sample and all samples in the training set using distance metrics such as Euclidean, Manhattan, or Minkowski distance. The k closest neighbors are selected, and the voting method (counting the frequency of each category in the k neighbors) is used to determine the sample’s category.

Linear Discriminant Analysis (LDA). LDA is used to extract features and classify spectral data by maximizing the separability between classes. Based on Bayesian theory, LDA calculates the intra-class scatter matrix, which measures the dispersion of samples within the same class, and the inter-class scatter matrix, which reflects the differences between classes. By solving a generalized eigenvalue problem, LDA finds the optimal projection direction, which is then used to project the data into a lower-dimensional space for classification.

Convolutional Neural Network (CNN). CNN was employed to classify the hyperspectral data of *A. roxburghii* and its varieties. The model consists of three 1D convolutional layers, each followed by max-pooling, to extract and downsample features from the spectral data. The convolutional layers use ReLU activation and L1 regularization to prevent overfitting. After feature extraction, the data is flattened and passed through a fully connected layer with 128 neurons, followed by a softmax output layer with 9 units corresponding to the 9 plant categories, as shown in Figure 2. The model is compiled with the Adam optimizer and sparse categorical cross-entropy loss and trained for 500 epochs. This architecture allows the model to learn complex patterns in the hyperspectral data, achieving accurate classification of different varieties of *A. roxburghii*.

## 3. Results and Discussion

### 3.1. Authenticity Identification

The initial spectral curves of the nine *A. roxburghii* varieties and their counterfeit species Figure 3a,b exhibited high similarity across most wavelengths, making visual differentiation challenging. To address this, machine learning algorithms were employed to extract discriminative spectral features. Among the tested models, the Support Vector Machine (SVM) demonstrated exceptional performance in distinguishing *A. roxburghii* from its counterfeit counterparts. The SVM model was applied to spectral data from both the adaxial (front) and abaxial (back) leaf surfaces, achieving 100% classification accuracy across all authenticity identification tasks. The confusion matrices Figure 3c,d displayed perfect classification with no misclassifications, confirming the model’s ability to accurately distinguish between *A. roxburghii* and counterfeit species. The confusion matrix presents results from the test set, which consists of 20% of the total samples per class. Given that each variety contains 10 leaves, this results in two test samples per class, leading to values such as 8/8 instead of 10/10. Specifically, the matrices for both front and back leaf classifications showed a diagonal structure (e.g., [8 0], [0 8]), indicating zero misclassifications in all cases. To assess the potential risk of overfitting [26], a learning curve analysis was conducted in Figure 3e. The figure illustrates model accuracy across different training sample sizes, with the training set accuracy shown in blue and the validation set accuracy in red. As the number of training samples increases, both accuracies exhibit a stable trend with minimal discrepancy. Notably, when using 35 samples, the accuracy difference between the training and validation sets is less than 0.04. Furthermore, once the training sample size reaches 50, both accuracies converge to 1, indicating consistent model performance across training and validation sets without signs of overfitting. These results demonstrate that both training and validation losses converge smoothly, confirming the model’s strong generalization capability.

From a mechanistic perspective, the SVM’s high classification accuracy can be attributed to several key factors. The spectral differences between *A. roxburghii* and its counterfeit species are primarily driven by variations in chemical composition, surface texture, and light reflectance characteristics. These differences are reflected in the distinct spectral responses of the front and back leaf surfaces, which are influenced by factors such as chlorophyll content, cell arrangement, and surface morphology. The SVM was able to exploit these unique spectral characteristics, particularly in the near-infrared and visible regions, to create clear decision boundaries between the authentic and counterfeit samples. The superior performance of the SVM can be attributed to the following factors: (1) high-dimensional feature space [27]: SVM maps the input spectral data into a higher-dimensional space using a Radial Basis Function (RBF) kernel, which enables linear separation of classes that are non-linearly separable in the original space. This transformation effectively enhances the spectral differences between the authentic and counterfeit species, which are influenced by factors such as chemical composition, surface morphology, and light reflectance properties. (2) Margin Maximization [28]: A key feature of SVM is its ability to maximize the margin between classes, which improves classification accuracy by ensuring robust generalization. This mechanism reduces overfitting, which is particularly important in hyperspectral data, where high dimensionality and limited sample sizes can lead to overfitting issues. Additionally, SVM’s inherent robustness to noise and outliers contributed to stable performance [29], even in the presence of minor spectral fluctuations. These results underscore the suitability of SVM for hyperspectral authenticity identification, particularly when leveraging multi-view spectral data. The model’s ability to effectively distinguish between *A. roxburghii* and its counterfeit species highlights its potential as a powerful and non-destructive tool for the authentication of herbal medicines.

### 3.2. Traditional Machine Learning Models for Goldthread Classification

Traditional machine learning methods, such as SVM, KNN, and LDA, have been widely applied in hyperspectral plant species identification. However, the performance of these models is often heavily influenced by data feature selection and preprocessing methods. Data preprocessing is particularly crucial in hyperspectral data, as it is often affected by noise and incomplete data. This section evaluates the classification performance of these traditional models when applied to the classification of nine varieties of *A. roxburghii* using different preprocessing methods, including MF, AF, GF, S-G Filtering, and PCA. To comprehensively assess the performance of these models, accuracy, precision, recall, and F1-score were used as performance metrics. These metrics evaluate the overall classification performance of the models, with accuracy measuring the proportion of correct predictions, precision indicating the reliability of positive predictions, recall reflecting the model’s ability to identify relevant instances, and F1-score providing a balanced measure of precision and recall, as shown in Figure 4.

Despite using a range of preprocessing methods and model combinations, the classification accuracy was generally low, with little improvement in performance. Among the tested models, SVM performed the best when classifying the back leaf surface of *A. roxburghii*, achieving an accuracy of around 0.8 for all four metrics when preprocessing with either AF or S-GSG. However, KNN performed poorly, with most metrics below 0.5, and showed no significant improvement with different preprocessing methods. This is likely due to KNN’s sensitivity to local fluctuations in data; when the distribution of the data is relatively simple and there are no major variations, preprocessing has a minimal effect on the results. LDA showed a significant dependence on preprocessing, but overall, its performance was not high. Accuracy for the front leaf was around 0.5, and for the back leaf, it was around 0.7, with the best performance of 0.8 achieved using GF. These results indicate that traditional machine learning models, such as SVM, KNN, and LDA, in combination with various preprocessing methods, did not achieve the desired classification accuracy. This suggests that traditional machine learning models struggle with the complexity of hyperspectral data, especially when classifying fine spectral differences between *A. roxburghii* and its counterfeit species. The relatively low performance could be attributed to the difficulty these models face in handling high-dimensional, noisy data, as well as the inherent limitations of hand-crafted feature extraction techniques. Therefore, there is a need to explore more effective or advanced classification models that can better handle the complexity of hyperspectral data and improve classification accuracy.

### 3.3. Multi-View Spectral Fusion Model

We further employed a Convolutional Neural Network (CNN) to classify the nine varieties of *A. roxburghii*. Figure 5 presents the training results of the model. (a) shows the loss function, where the sparse categorical cross-entropy loss was used, which is suitable for multi-class classification problems, especially when the labels are encoded as integers. The formula for the loss function is as follows [30]:$$L=-\sum_{i=1}^{C}y_{i}\log(p_{i})$$
where *L* is the loss value, *C* is the total number of classes, *y_i_* is the true label of class *i* (1 if the sample belongs to class *i*, otherwise 0), and *P_i_* is the model’s predicted probability for class *i*. Figure 5a shows the change in the loss function during training. It is evident that as the number of training epochs increases, the loss value decreases significantly and stabilizes, indicating an effective learning process of the model. Figure 5b illustrates the accuracy trend during training. As seen in the figure, the accuracy steadily approaches 1 as training progresses, with reduced fluctuation in the accuracy, further confirming the stability and reliability of the model. Figure 5c displays the confusion matrix for the trained model on the test set. The matrix reveals that the model correctly classified all samples for each category with no misclassifications. Table 1 provides a comprehensive performance comparison of all models (SVM, KNN, LDA, and CNN) in terms of accuracy, precision, recall, and F1-score. This table improves result transparency and effectively highlights the CNN model’s superiority over traditional machine learning approaches. To further evaluate the reliability of our model, we computed the standard deviation and 95% confidence interval (CI) for the test set. Since spectral data classification does not inherently involve standard deviations for test set accuracy, we instead report the standard deviation and 95% CI of training set accuracy over multiple runs. Table 1 presents the test accuracy alongside the mean and standard deviation of training accuracy, demonstrating the stability of model performance. The CNN model consistently achieved 100% accuracy, while traditional machine learning models exhibited variability. The 95% CI values indicate that our training process is stable, minimizing concerns about overfitting. This result demonstrates the exceptional performance of the multi-view spectral fusion model, which successfully utilized spectral data from both the adaxial (test accuracy: 0.8889) and abaxial (test accuracy: 0.9722) leaf surfaces to improve classification accuracy.

To assess whether the performance differences among the models are statistically significant, we conducted a one-way ANOVA test on the accuracy results of all nine models. The analysis yielded an F-statistic of 298.59 and a *p*-value of approximately 9.03 × 10^−57^, indicating highly significant differences among model performances (*p* < 0.001). A post hoc Tukey HSD test was further applied to perform pairwise comparisons between models. The results, visualized in Figure 5d, reveal that the CNN-based models (CNN-Front, CNN-Back, CNN-Fusion) significantly outperformed the traditional machine learning models (SVM, KNN, LDA), providing robust statistical evidence of the effectiveness of deep learning in this classification task. As shown in Figure 5c, the 100% accuracy can be attributed to several factors: complementarity, data augmentation, and feature diversity within the model’s structure and optimization. By using spectral data from both the front and back leaf surfaces, the model leverages multi-view information, providing richer feature representations. The spectral responses of the front and back leaf surfaces may differ in certain physical properties, such as light reflectance and scattering characteristics. These differences help capture the varying structures and compositions of the leaves, ultimately improving classification precision. The spectral data from the front and back surfaces are complementary, enhancing the robustness and accuracy of the model. Moreover, different leaf species exhibit significant spectral differences, particularly in specific wavelength regions. The front and back spectra provide different angles of information, which can help better capture these subtle spectral differences. By incorporating this additional feature dimension (i.e., both front and back spectra), the model gains more information, thus improving its generalization ability. The introduction of feature diversity allows the model to identify more potential patterns during training, thereby enhancing classification accuracy. In contrast to traditional hand-crafted feature extraction methods, deep learning models like CNNs can automatically learn optimal feature combinations through end-to-end training, resulting in more precise classifications. As the number of layers and neurons in the network increases, the model can process more complex spectral data and extract deeper-level features. Overall, the qualitative model developed in this study demonstrated excellent training performance and generalization ability. These results highlight the potential of CNN-based models in hyperspectral data classification, particularly when leveraging multi-view spectral fusion. The model’s ability to effectively capture the fine spectral differences between *A. roxburghii* varieties underscores the effectiveness of deep learning techniques in tackling complex classification tasks in the realm of plant species identification. Our findings suggest that even with a limited number of samples, HSI combined with machine learning can effectively classify *A. roxburghii* varieties and detect adulterants. This highlights the potential of small-sample learning approaches in spectral analysis, particularly for rare medicinal plants where large-scale data collection is challenging.

## 4. Conclusions

This study successfully demonstrated the application of hyperspectral imaging and machine learning techniques for the high-precision classification and authenticity identification of *Anoectochilus roxburghii* and its counterfeit species. Among the machine learning models tested, the SVM model showed exceptional performance, achieving 100% accuracy in distinguishing Goldthread from its counterfeit species by leveraging spectral data from both the front and back leaves. Traditional machine learning models, such as KNN and LDA, exhibited relatively lower classification accuracy, especially when applied to complex hyperspectral data, highlighting the limitations of these models in handling high-dimensional features. The introduction of the multi-view spectral fusion model, which combines the spectral data from both the front and back sides of the leaves, significantly improved classification performance and robustness, demonstrating the benefits of utilizing complementary information for classification tasks. The findings of this study offer valuable insights into the potential applications of hyperspectral imaging and machine learning for authenticity identification and counterfeit detection in medicinal plants, with implications for improving quality control and preventing fraud in the herbal medicine industry. Furthermore, the proposed multi-view fusion model provides a promising approach for improving the accuracy of plant species identification, which could be extended to a wide range of botanical applications. In addition, these results contribute to the broader ecological and economic context by promoting sustainable use of genuine herbal resources, protecting endangered medicinal plant species from overexploitation, and supporting fair market practices through reliable authentication technologies. Such advancements are crucial for maintaining biodiversity, ensuring consumer safety, and enhancing the integrity of the herbal medicine supply chain.

## Figures and Tables

**Figure 1 plants-14-01177-f001:**
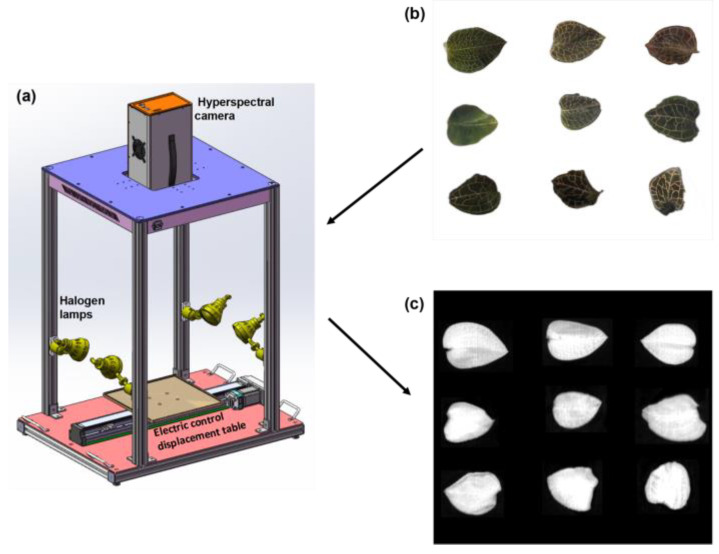
(**a**) Schematic diagram of the hyperspectral experimental setup; (**b**) photographs of the 9 varieties of *A. roxburghii* samples; (**c**) hyperspectral images of the samples.

**Figure 2 plants-14-01177-f002:**
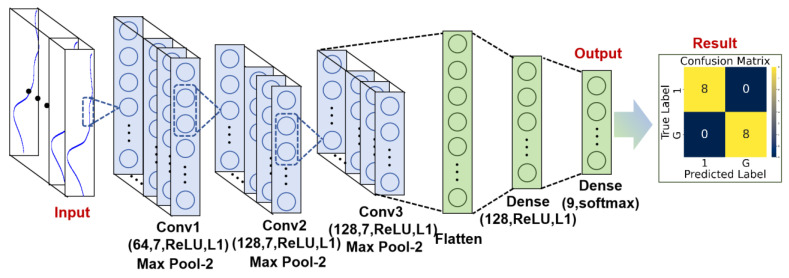
Architecture of the CNN model used for classifying hyperspectral data of *A. roxburghii* varieties.

**Figure 3 plants-14-01177-f003:**
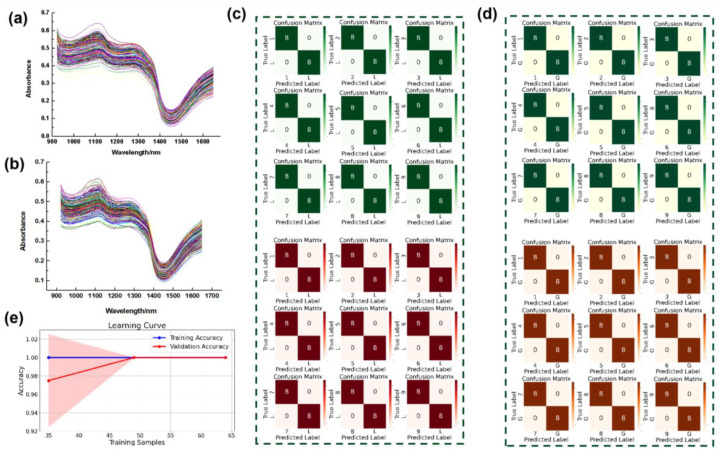
(**a**,**b**) Original hyperspectral spectra of the adaxial (front) and abaxial (back) surfaces of *A. roxburghii*; (**c**,**d**) confusion matrices for the classification of *A. roxburghii* (adaxial and abaxial) versus counterfeit species (adaxial and abaxial). (**e**) Learning curve. The blue line represents training accuracy, and the red line represents validation accuracy across different training sample sizes.

**Figure 4 plants-14-01177-f004:**
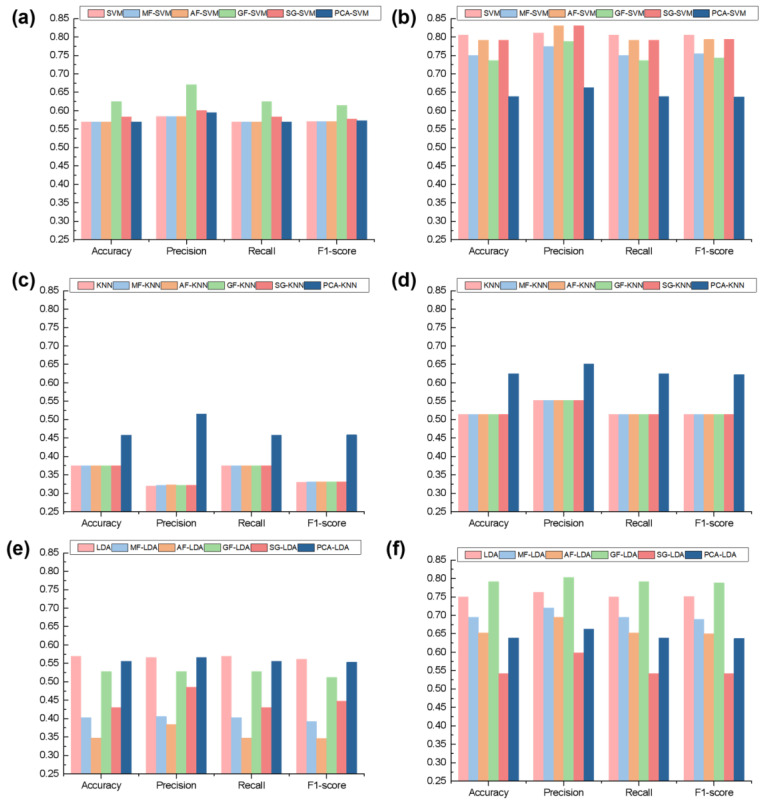
Classification results of *A. roxburghii* using SVM, KNN, and LDA. (**a**,**c**,**e**) represent the classification results for the adaxial (front) leaf surface; (**b**,**d**,**f**) show the classification results for the abaxial (back) leaf surface.

**Figure 5 plants-14-01177-f005:**
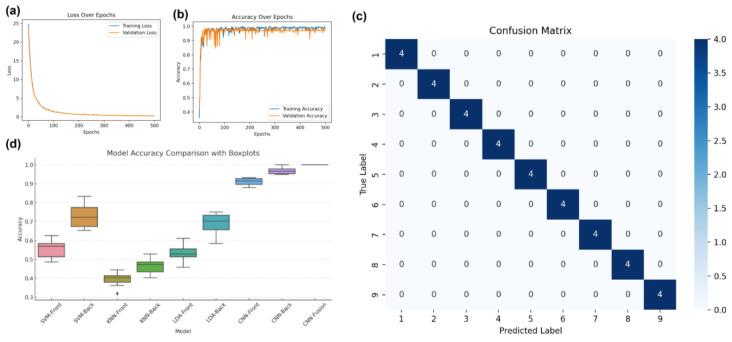
Training results of the CNN model. (**a**) Loss function during training. (**b**) Accuracy progression during training. (**c**) Confusion matrix of the model’s classification performance on the test set. (**d**) Results of the Tukey HSD test comparing model classification accuracy.

**Table 1 plants-14-01177-t001:** Comparison of classification performance across different models.

Model		Accuracy	Precision	Recall	F1-Score	Train Accuracy	95% CI (Train Accuracy)
SVM	front	0.5694	0.5844	0.5694	0.5704	0.5243 ± 0.0474	[0.4584, 0.5901]
	back	0.8056	0.8113	0.8056	0.8055	0.6665 ± 0.0276	[0.6281, 0.7048]
MF-SVM	front	0.5694	0.5844	0.5694	0.5704	0.5383 ± 0.0404	[0.4881, 0.5885]
	back	0.7500	0.7738	0.7500	0.7545	0.6629 ± 0.0350	[0.6195, 0.7063]
AF-SVM	front	0.5694	0.5844	0.5694	0.5704	0.5382 ± 0.0388	[0.4843, 0.5920]
	back	0.7917	0.8304	0.7917	0.7936	0.6699 ± 0.0380	[0.6171, 0.7227]
GF-SVM	front	0.6250	0.6713	0.6250	0.6147	0.5416 ± 0.0263	[0.5089, 0.5743]
	back	0.7361	0.7882	0.7361	0.7428	0.6700 ± 0.0213	[0.6436, 0.6964]
SG-SVM	front	0.5833	0.6001	0.5833	0.5782	0.5415 ± 0.0343	[0.5115, 0.5715]
	back	0.7917	0.8304	0.7917	0.7936	0.6699 ± 0.0380	[0.6366, 0.7033]
PCA-SVM	front	0.5694	0.5953	0.5694	0.5731	0.5070 ± 0.0344	[0.4643, 0.5497]
	back	0.6389	0.6624	0.6389	0.6371	0.6495 ± 0.0275	[0.6153, 0.6837]
KNN	front	0.3750	0.3199	0.3750	0.3302	0.4056 ± 0.0893	[0.2946, 0.5165]
	back	0.5139	0.5528	0.5139	0.5144	0.4750 ± 0.0753	[0.3816, 0.5684]
MF-KNN	front	0.3750	0.3215	0.3750	0.3310	0.4056 ± 0.0972	[0.2849, 0.5262]
	back	0.5139	0.5528	0.5139	0.5144	0.4806 ± 0.0759	[0.3864, 0.5748]
AF-KNN	front	0.3750	0.3235	0.3750	0.3319	0.4586 ± 0.0359	[0.4140, 0.5032]
	back	0.5139	0.5528	0.5139	0.5144	0.4615 ± 0.0653	[0.3805, 0.5426]
GF-KNN	front	0.3750	0.3215	0.3750	0.3310	0.4056 ± 0.0972	[0.2849, 0.5262]
	back	0.5139	0.5528	0.5139	0.5144	0.4806 ± 0.0759	[0.3864, 0.5748]
SG-KNN	front	0.3750	0.3215	0.3750	0.3310	0.4517 ± 0.0380	[0.4184, 0.4850]
	back	0.5139	0.5528	0.5139	0.5144	0.4615 ± 0.0653	[0.4043, 0.5187]
PCA-KNN	front	0.4583	0.5152	0.4583	0.4585	0.4064 ± 0.0655	[0.3489, 0.4638]
	back	0.6250	0.6515	0.6250	0.6224	0.5694 ± 0.0359	[0.5379, 0.6008]
LDA	front	0.5694	0.5665	0.5694	0.5609	0.5139 ± 0.1185	[0.3667, 0.6610]
	back	0.7500	0.7625	0.7500	0.7517	0.6833 ± 0.0671	[0.6000, 0.7667]
MF-LDA	front	0.4028	0.4059	0.4028	0.3923	0.4410 ± 0.0817	[0.3694, 0.5125]
	back	0.6944	0.7205	0.6944	0.6893	0.7044 ± 0.0734	[0.6401, 0.7688]
AF-LDA	front	0.3472	0.3847	0.3472	0.3464	0.4417 ± 0.0397	[0.3924, 0.4909]
	back	0.6528	0.6943	0.6528	0.6500	0.6194 ± 0.0509	[0.5562, 0.6827]
GF-LDA	front	0.5278	0.5284	0.5278	0.5125	0.4621 ± 0.1120	[0.3231, 0.6012]
	back	0.7917	0.8037	0.7917	0.7878	0.6909 ± 0.0589	[0.6178, 0.7640]
SG-LDA	front	0.4306	0.4858	0.4306	0.4470	0.4204 ± 0.0864	[0.3005, 0.5404]
	back	0.5417	0.5983	0.5417	0.5424	0.6245 ± 0.0620	[0.5385, 0.7105]
PCA-LDA	front	0.5556	0.5663	0.5556	0.5532	0.5106 ± 0.0654	[0.4532, 0.5680]
	back	0.6389	0.6629	0.6389	0.6372	0.6075 ± 0.0435	[0.5694, 0.6456]
CNN	front	0.9028	0.9387	0.9028	0.9039	0.9531 ± 0.0064	[0.9452, 0.9610]
	back	0.9722	0.9773	0.9722	0.9690	0.9939 ± 0.0052	[0.9874, 1.0000]
**CNN**	**fusion**	**1.0000**	**1.0000**	**1.0000**	**1.0000**	**1.0000** **±** **0**	**[1.0000, 1.0000]**

## Data Availability

A subset of the dataset, including representative hyperspectral images and classification labels, has been made publicly available at [10.6084/m9.figshare.28711589]. The full dataset can be accessed upon reasonable request.

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
