# Peer review of "Small-Sample Authenticity Identification and Variety Classification of Anoectochilus roxburghii (Wall.) Lindl. Using Hyperspectral Imaging and Machine Learning"

_plants, 2025, doi:10.3390/plants14081177_

Round 1

Reviewer 1 Report

Comments and Suggestions for Authors

A short article, written in a clear manner. Well-defined aim. The article fits the profile of the magazine.
On the other hand, changes are needed before acceptance.

Please write what other methods are used to analyze medicinal plant materials.
What are the advantages of Hyperspectral Image Analysis and Machine Learning Techniques. compared to other techniques
What are the disadvantages and limitations of the method used.
Please mention where else this technique is applied in botany. For example:

https://www.mdpi.com/2071-1050/16/14/6064

Reviewer 2 Report

Comments and Suggestions for Authors

Dear Authors,

Dear Editors,

The manuscript applies a modern technological approach (hyperspectral imaging – HSI, and machine learning) to identify varieties of Anoectochilus roxburghii (golden root) and detect counterfeits. The topic is relevant, particularly in the context of the traditional medicine market. The strength of the study lies in the use of multi-view HSI and a CNN-based model, which demonstrates remarkable accuracy. However, several critical aspects need to be improved to achieve publication readiness.

The true innovation of the study is not sufficiently highlighted in comparison to similar works – previous studies (e.g., Li et al., 2018; Chai et al., 2021) have also combined NIR and CNN approaches. The actual advantages of the multi-view analysis based on HSI should be compared more thoroughly with past methods.

Further serious issues include: the sample size is too small (10 leaves/variety) → it does not allow for robust validation of the model. There is no information about data availability (only “on request”) – open data sharing would be necessary for publication. The results of preprocessing variants are not reproducible, as parameters (e.g., SG-filter window size) are not provided. Too many models were used by the authors – SVM, KNN, LDA, and CNN are all mentioned, but a comprehensive comparison table of all models and metrics is missing. The reported 100% accuracy seems suspicious, and the possibility of overfitting is not discussed. It is unclear how data separation (training/test) was performed, and whether any real external validation exists. The confusion matrices are not entirely interpretable (e.g., why 8/8 samples instead of 10/10?). Statistical significance or error margin analysis is missing. The figures (visual materials) are not accessible, making it impossible to visually verify the results.

The manuscript is generally understandable, but I would point out some editorial and stylistic issues: there are repetitions in several places; numbering of sections is inconsistent. The English language is generally clear, but some terminology is overly complex. There are 21 references, most of which are appropriate; however, some are missing or outdated (e.g., SSLChina: IFWS 2020 is difficult to retrieve), and these should be replaced.

The topic of the article is relevant, but it requires significant methodological and formal improvements, especially concerning data volume, model validation, and transparency of results.

Comments on the Quality of English Language

The manuscript is generally understandable, but I would point out some editorial and stylistic issues: there are repetitions in several places; numbering of sections is inconsistent. The English language is generally clear, but some terminology is overly complex.

Reviewer 3 Report

Comments and Suggestions for Authors The article is interesting because of the objective pursued with a species of taxonomic importance and in the field of medicinal botany. However, the following remark should be borne in mind:

With regard to the training and validation of machine learning models, it is not entirely clear how the validation of the models has been carried out. The tuning process of the hyperparameters has been carried out by means of cross-validation. However, there is no mention of a test set in the material and methods, which is mentioned in the results. In case the test sets refer to those used during the hyperparameter tuning phase of the cross-validation, this has been found in the literature to lead to model overfitting and loss of generalisability to new datasets (Bischl et al., 2023). It would therefore be necessary to specify further which test sets are referred to and have been used in the validation and for the calculation of the performance metrics of the model.

Bischl, B., Binder, M., Lang, M., Pielok, T., Richter, J., Coors, S., Thomas, J., Ullmann, T., Becker, M., Boulesteix, A.-L., Deng, D., & Lindauer, M. (2023). Hyperparameter optimization: Foundations, algorithms, best practices, and open challenges. WIREs Data Mining and Knowledge Discovery, 13(2), e1484. https://doi.org/10.1002/widm.1484 

Reviewer 4 Report

Comments and Suggestions for Authors

This manuscript describes the novel discrimination of Goldthread from two counterfeit species using hyperspectral imaging and several machine learning approaches.  Support vector machine was demonstrated to provide the most satisfactory performance.  A novel multi-view spectral fusion model for the front and back of the leaves provided enhanced accuracy.  This important and carefully performed study illustrates the use of hyperspectral imaging and machine learning for the characterization of medicinal plants.  The quality of the English is high and does not require improvement.

The novelty of this work focuses upon the combination of hyperspectral imaging with machine learning to accurately classify medicinal plants.  I anticipate this approach may also have applications for other plant systems, providing further significance.

The introduction is well written, provides suitable background, and incorporates the relevant literature.  The research design was appropriate and the methods were carefully described.  The authors clearly presented the results which supported the conclusions.

I have no conflicts of interest in reviewing this manuscript.  I saw no evidence of plagiarism and did not observe inappropriate self-citation.  In summary, I have no ethical concerns.

I believe the novelty of this work is average, as there are numerous studies of this type combining spectroscopy and machine learning to classify medicinal plants.  However, this application is clearly significant and scientifically sound.  The presentation and organization of the manuscript are very good and do not require significant revision.  I believe this study will be of interest to the Plants audience and conclude that the overall merit is high.

I therefore recommend acceptance of the manuscript in its current form.

Author Response

We sincerely appreciate your positive feedback and recognition of our research. We are delighted to hear that you acknowledge the novelty of our study in discriminating Anoectochilus roxburghii from its counterfeit species using hyperspectral imaging and machine learning. Your comments reinforce our confidence in this research direction and encourage us to further explore its potential applications in other plant systems.

We also appreciate your recognition of the manuscript’s writing quality, research design, methodological clarity, and presentation of results. Your valuable feedback is greatly encouraging, and we are pleased to have the opportunity to share our findings with the Plants audience.

Once again, thank you for your thoughtful review and recommendation for acceptance!

Round 2

Reviewer 2 Report

Comments and Suggestions for Authors

Dear Authors,

The revised manuscript has undergone substantial improvements that strengthen its scientific validity, methodological precision, and applicability. Although the results are convincing, it would be useful to perform a more detailed statistical comparison, for example by using a robust statistical test to detect differences between the individual models. Furthermore, it would be worthwhile to better emphasize how the results fit into the broader ecological and economic context of herbal authentication.

Reviewer 3 Report

Comments and Suggestions for Authors

Dear authors, thank you for considering the suggestions made in the review. I believe that the article has become more complete and reinforces the interest of the article. Consequently, I am re-evaluating the article and recommending its publication.
Best regards. 

Author Response

Response:
Thank you very much for your kind and encouraging feedback. We sincerely appreciate your time and effort in reviewing our manuscript. Your positive evaluation and recommendation for publication mean a great deal to us and reinforce our confidence in the value of this work.

Best regards.